# Oxidation of fish oil exacerbates alcoholic liver disease by enhancing intestinal dysbiosis in mice

Ruibing Feng[1,2], Li-Juan Ma[1], Meng Wang[1,3], Conghui Liu[1], Rujie Yang[1], Huanxing Su[1], Yan Yang[4] & Jian-Bo Wan [1][✉]

The role of n-3 polyunsaturated fatty acids (PUFAs) in alcoholic liver disease (ALD) has been controversial. N-3 PUFA oxidation in animal feeding stuffs was rarely concerned, likely contributing to inconsistent outcomes. Here, we report the impacts of oxidized fish oil (OFO) on ALD in C57BL/6 mice. Alcohol exposure increased plasma aminotransferase levels and hepatic inflammation. These deleterious effects were ameliorated by unoxidized FO but exacerbated by OFO. Sequencing analysis showed the accentuated intestinal dysbiosis and the increased proportion of *Proteobacteria* in OFO-fed mice. Intestinal sterilization by antibiotics completely abolished OFO-aggravated liver injury. Additionally, alcohol exposure leads to the greater increase in plasma endotoxin and decrease in intestinal tight junction protein expressions in OFO-fed mice. Stabilization of intestinal barrier by obeticholic acid markedly blunted OFO-aggravated liver injury in alcohol-fed mice. These results demonstrate that OFO exacerbates alcoholic liver injury through enhancing intestinal dysbiosis, barrier dysfunction, and hepatic inflammation mediated by gut-derived endotoxin.

[1] State Key Laboratory of Quality Research in Chinese Medicine, Institute of Chinese Medical Sciences, University of Macau, Macao, China. [2] Key Laboratory of Tropical Biological Resources of Ministry of Education, Key Laboratory for Marine Drugs of Haikou, School of Life and Pharmaceutical Sciences, Hainan University, Haikou 570228, China. [3] Center for Drug Innovation and Discovery, College of Life Science, Hebei Normal University, Shijiazhuang, Hebei, China. [4] Guangdong Provincial Key Laboratory of Food, Nutrition and Health, School of Public Health, Sun Yat-sen University, Guangzhou, China. [✉]email: jbwan@um.edu.mo

Prolonged and excessive amount of alcohol consumption causes alcoholic liver disease (ALD), ranging from simple hepatosteatosis, hepatitis, hepatic fibrosis, and cirrhosis[1,2]. In 2016, alcohol abuse was responsible for ~3 million deaths and 132.6 million disability-adjusted life years worldwide[3]. The critical role of gut–liver axis has been well established both in clinical and experimental models of ALD[4]. Alcohol exposure disturbs gut microbiota homeostasis by alteration in bacteria composition, particularly the overgrowth of pathogenic bacteria. Accumulating evidence indicated that gut microbiota manipulation, such as fecal microbial transplantation and the treatment of probiotics or antibiotics, ameliorates alcoholic liver injury in humans and rodent models[4,5]. The overgrowth of intestinal microbiota, especially Gram-negative, increases the bacterial endotoxin or lipopolysaccharides (LPS). Alcohol exposure also impairs intestinal barrier integrity, leading to the increased intestinal permeability, and subsequent the enhanced translocation of bacteria and LPS to the portal circulation[6]. Clinical and experimental studies have indicated that the circulating level of LPS is indeed correlated with the severity of alcoholic liver injury[7,8]. The gut-derived LPS activates hepatic Kupffer cells by Toll-like receptor 4 (TLR4)-mediated nuclear factor κB (NF-κB) signaling, resulting in the overproduction of inflammatory cytokines, which, in turn, induces hepatocyte damage[9].

Intestinal barrier dysfunction is a major cause of alcohol-induced bacterial translocation, and has been considered as a prerequisite in the development of ALD[10,11]. Although the exact molecular mechanisms underlying intestinal barrier dysfunction in the pathogenesis of ALD are not sufficiently elucidated. Acetaldehyde, a metabolite of alcohol, is correlated with the disruption of intestinal integrity by tyrosine phosphorylation of intestinal epithelial tight junction (TJ) proteins[12]. TJ proteins are consisted of a variety of transmembrane proteins, including intracellular plaque protein zonula occludin (ZO-1), occludin, claudin-2, and caudin-4[13]. Accumulating evidence has shown that dysbiosis-caused intestinal inflammation also contributed to the increased intestinal permeability in ALD[14,15]. TNF-α, a well-characterized mediator of inflammatory response, is increased in the jejunum of mice with enteric dysbiosis, and has demonstrated to be associated with increasing intestinal permeability after alcohol feeding[16].

In recent decades, long-chain n-3 polyunsaturated fatty acids (PUFAs), including eicosapentaenoic acid (EPA, 20:5 n-3) and docosahexaenoic acid (DHA, 22:6 n-3), have been well documented to exert the potentially therapeutic efficacy in nonalcoholic liver disease[17], and liver injury induced by various chemicals, including carbon tetrachloride[18], acetaminophen[19], concanavalin A[20], and D-galactosamine/lipopolysaccharide[21]. An increasing evidence has demonstrated that n-3 PUFAs exerted the beneficial effects on ALD[9,22–24]. However, the role of n-3 PUFAs in ALD has been still inconsistent and controversial. Several studies have shown that the supplement of n-3-rich fish oil (FO) exacerbates alcoholic liver injury and promotes the pathogenesis of ALD, which is attributed to the intensified oxidative stress and inflammation in the liver[25–28]. But it is well known that n-3 PUFAs with high degree of unsaturation are highly susceptible to oxidation and exert antioxidative and anti-inflammatory properties. It remains unclear whether the high levels of oxidative stress and inflammation observed in these studies with opposite results are caused by n-3 PUFAs or their oxidation products already existing in FO used[2,29].

Therefore, the aim of the present study is to investigate the probable impacts of the oxidized fish oil (OFO) on the pathogenesis of ALD. Here we show that oxidation of FO exacerbates alcoholic liver injury through enhancing intestinal dysbiosis, barrier dysfunction, and hepatic inflammation mediated by gut-derived endotoxin.

## Results

**OFO exacerbates the features of ethanol-induced liver injury.** The primary and secondary lipid-oxidation products were measured to determine oxidation status in FO and OFO. After air exposure at 65 °C for 2 weeks, OFO showed the higher degree of oxidation as evidence by the greatly elevated levels of peroxide value (POV), p-anisidine value (AV), total oxidation value (Totox), and thiobarbituric acid-reactive substances (TBARS), compared to the unoxidized FO (Table 1). As determined by gas chromatography–mass spectrometry (GC-MS), the contents of n-3 PUFAs, including EPA, DPA and DHA, in OFO were greatly decreased, compared to unoxidized FO. Approximately 50.9% EPA and 58.7% DHA in FO were oxidized during the process (Supplementary Table 1 and Supplementary Fig. 1). Liver injury induced by ethanol exposure was evaluated by biochemical and histological analysis, and oxidative stress in liver. Acute-on-chronic alcohol feeding (Supplementary Fig. 2) elevated plasma levels of alanine aminotransferase (ALT) and aspartate aminotransferase (AST) by 4.5-fold and 3.0-fold, respectively, in CO-treated mice. These elevations were greatly augmented to 8.1-fold and 4.6-fold, respectively, by replacing a half of CO with OFO, but significantly decreased by supplement with FO (Fig. 1a). OFO-treated mice had a pronounced increase in plasma triglyceride (TG) and hepatic TG levels, compared to the CO-treated mice after alcohol exposure (Fig. 1b). Alcohol feeding caused oxidative stress in the liver from both CO- and OFO-treated mice, as characterized by the decreased antioxidant parameters (e.g., CAT, SOD, GSH), and increased the hepatic malonaldehyde (MDA) level, but the effects were much greater in OFO-treated mice (Fig. 1c). Lipid accumulation in the liver was increased by alcohol exposure in CO-treated mice, and markedly increased in OFO-treated group, but remarkably decreased by FO treatment, as measured by hematoxylin and eosin (H&E) (Fig. 1d) and oil red O staining (Supplementary Fig. 3a). Taken together, these results indicate that hepatic steatosis and oxidative stress induced by alcohol exposure are exacerbated by dietary OFO but reversed by FO.

To determine whether the absorption and metabolism of ethanol is implicated in these deteriorative effects of OFO, the circulating ethanol concentration and hepatic expression of cytochrome P450 2E1 (CYPE21) were measured. After chronic-plus-single-binge ethanol feeding, the ethanol concentrations in plasma were greatly elevated in both CO- and OFO-treated mice. However, plasma ethanol levels were comparable between CO and OFO groups after ethanol exposure (Supplementary Fig. 4a). Similar tendency was observed for hepatic expression of CYPE21 (Supplementary Fig. 4b).

**OFO aggravates ethanol-induced hepatic inflammation.** Alcohol exposure increased the hepatic inflammation in both CO- and OFO-treated mice, as shown by the increased pro-inflammatory cytokines (TNF-α, IL-6, and IL-1β) and the

**Table 1 Measurement of the oxidized parameters in FO and OFO.**

| Parameters | FO | OFO |
|---|---|---|
| POV (meq kg$^{-1}$) | 1.06 ± 0.23 | 34.7 ± 3.2* |
| AV | 5.58 ± 0.57 | 40.4 ± 5.6* |
| Totox | 7.71 ± 0.76 | 109.9 ± 12.1* |
| TBARS (MDA, mg kg$^{-1}$) | 1.62 ± 0.46 | 58.9 ± 6.9* |

Values are expressed as means ± SD ($n = 3$); *$p < 0.05$.
OFO oxidized fish oil, FO fish oil, POV peroxide value, AV p-anisidine value, Totox total oxidation, TBARS the thiobarbituric acid-reactive substances, MDA malonaldehyde.

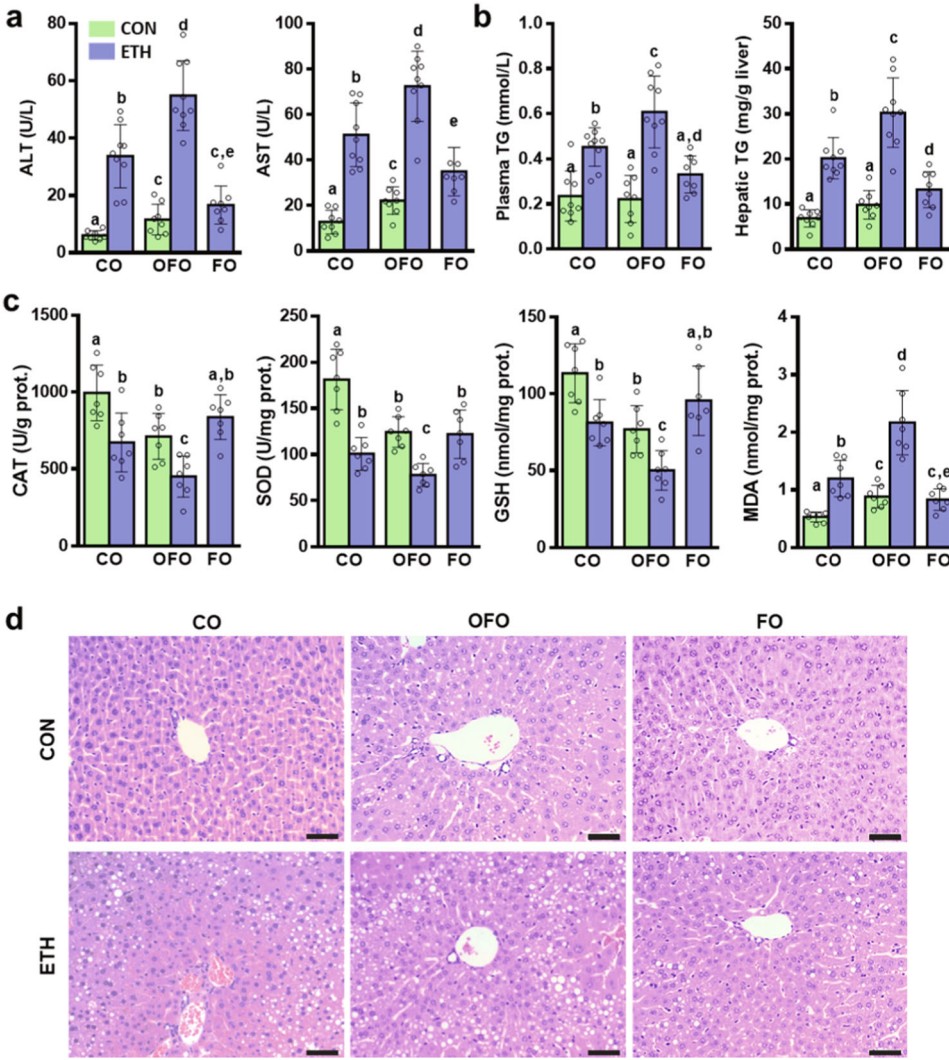

**Fig. 1 OFO exacerbates ethanol-induced liver injury in mice. a** Plasma levels of ALT and AST ($n = 8$–9). **b** Plasma TG level and hepatic TG concentration ($n = 8$–9). **c** Oxidative stress parameters in the liver, including CAT, SOD, GSH, and MDA ($n = 7$). **d** Representative H&E staining of liver sections (scale bar, 50 μm). Data were expressed as mean ± SD. Labeled means without a common letter differ within the column ($p < 0.05$). CO corn oil, OFO oxidized fish oil, FO fish oil.

chemokine (MCP-1), and the decreased anti-inflammatory cytokine (IL-10), but the effects were much greater in OFO-treated mice. While FO supplement decreased these pro-inflammatory factors induced by ethanol feeding, but did not rescue hepatic IL-10 (Fig. 2a, b), we speculated that the inhibition of pro-inflammatory factors, but not normalization of anti-inflammatory cytokines, contribute to the protective effects of FO against alcoholic hepatic inflammation in the mouse model of chronic-plus-single-binge ethanol feeding. Consistently, the immunofluorescent staining of F4/80, a specific monocyte/macrophage marker, showed that more F4/80-positive cell number was observed in the section of liver tissue from OFO-treated mice after alcohol feeding, compared to CO group. While FO decreased the macrophage infiltration in the liver (Supplementary Fig. 3b). These data suggest that dietary OFO exacerbates hepatic inflammation upon alcohol challenge.

Activation of Kupffer cell via TLR4/NF-κB signaling plays a vital role in hepatic inflammation induced by ethanol exposure[4,30]. Next, we examined the protein expressions of the genes related to TLR4/NF-κB signaling in the liver. As shown in Fig. 2c, OFO-treated mice had a remarkable increase in hepatic expressions of TLR4, and its downstream MyD88 and

phosphorylated p65 (p-p65) without changing total p65 expression, compared to CO-treated mice after alcohol feeding. These increased protein expressions were reversed by supplement with FO. In addition, OFO alone mice showed slight increases in hepatic expressions of TLR4, MyD88, and p-p65, compared to CON/CO group (Fig. 2c). Consistently, the immunofluorescent staining revealed the higher hepatic expression of p-p65 in OFO-treated group after alcohol intake, compared to CO-treated mice (Fig. 2d). Collectively, our data demonstrate that TLR4/NF-κB signaling is implicated in OFO-exacerbated hepatic inflammation in alcohol-fed mice.

**OFO enhances ethanol-induced intestinal barrier dysfunction.** We next examined whether intestinal barrier integrity is influenced by dietary OFO. TJ proteins, the markers of intestinal integrity, play the crucial role in maintaining barrier function of intestinal epithelium[31,32]. Immunofluorescent and western blot analysis clearly showed the decreased expression of TJ proteins, including ZO-1, occludin, and claudin-4, in jejunum tissue from CO-treated mice after alcohol exposure, along with the increased expression of claudin-2, a TJ protein involving in the formation of

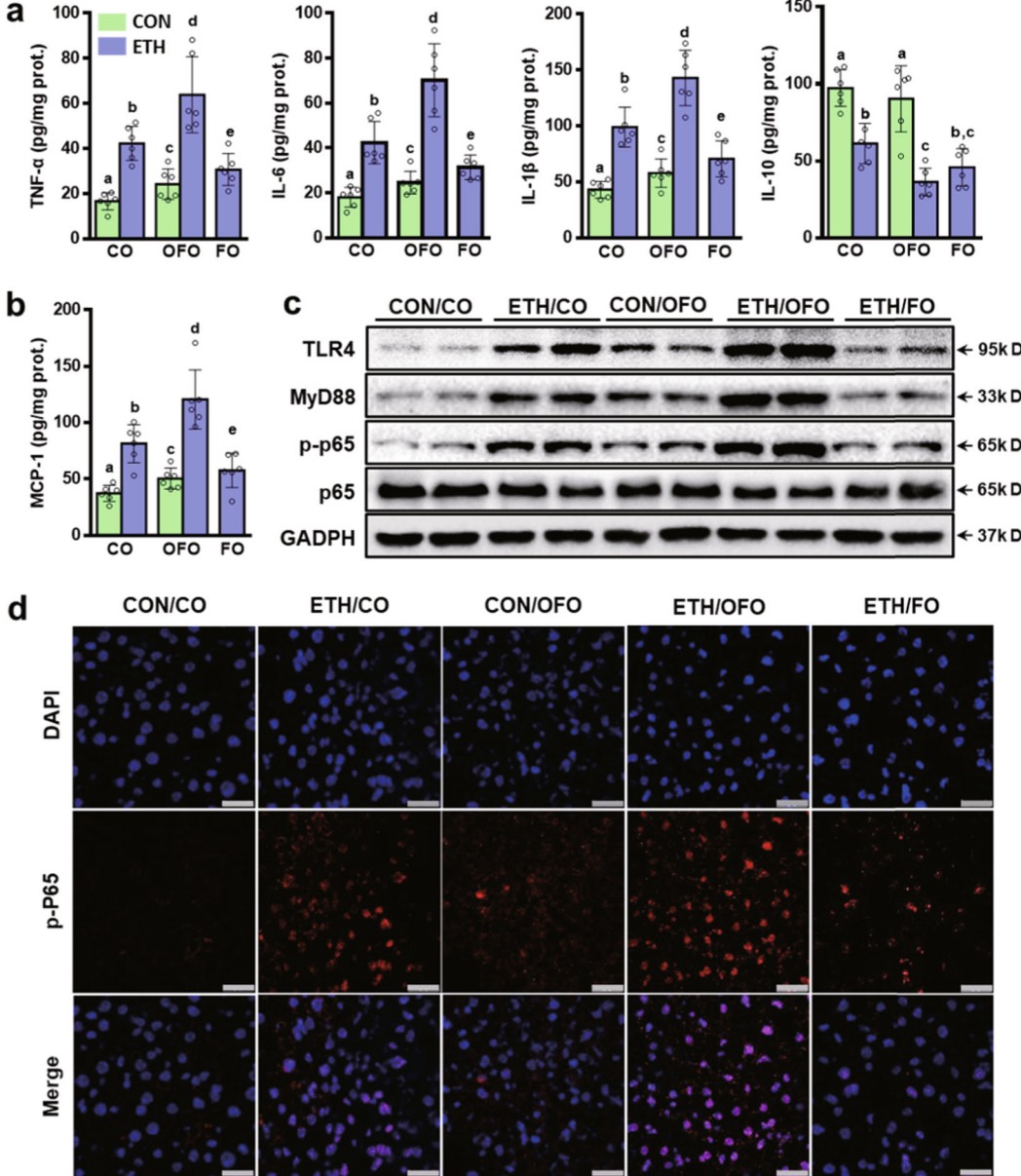

**Fig. 2 OFO aggravates ethanol-induced hepatic inflammation. a** Hepatic level of cytokines (TNF-α, IL-6, IL-1β, and IL-10). **b** chemokine (MCP-1).
**c** Immunoblot analysis of TLR4, MyD88, phosphorylated p65, and total p65 proteins in the liver. **d** Immunofluorescent staining of liver sections for the phosphorylated p65 (red), nucleus was stained with DAPI (blue; scale bar, 25 μm). Data were expressed as mean ± SD ($n = 6$). Labeled means without a common letter differ within the column ($p < 0.05$). CO corn oil, OFO oxidized fish oil, FO fish oil.

paracellular water channel that is typically expressed in leaky epithelial tissues. But these changes were more severe in OFO-treated mice, while were reversed by supplement with FO (Fig. 3a, b). Importantly, OFO alone did not affect the expressions of all TJ protein investigated, compared to CO alone group (Fig. 3a, b). The impaired intestinal barrier integrity leads to the increased intestinal permeability to bacterial products, such as LPS, and the greater bacteria translocation. After chronic-plus-single-binge ethanol feeding, plasma LPS level and the hepatic expression of gram-negative bacterial 16S rRNA, a marker of bacterial translocation, were increased in both CO- and OFO-treated mice, much greater increase was observed in OFO group, but significantly decreased by FO supplement. Although OFO alone group has the slightly increased hepatic expression of 16S rRNA, compared to CO alone group, but their plasma LPS levels were comparable (Fig. 3c, d). While fecal LPS contents were increased

in both CO- and OFO-treated mice, OFO alone mice showed the significantly increased LPS level in feces, compared to CO alone group (Fig. 3e). A most plausible explanation is that OFO alone cause the great increased fecal LPS, which may not be diffused into the circulation, because OFO alone cannot greatly disrupt the intestinal epithelial barrier.

An increasing evidence has demonstrated that intestinal inflammation contributed to intestinal barrier dysfunction[33]. To further investigate the role of OFO in intestinal inflammation in ethanol-fed mice, immunofluorescent staining of F4/80 and TNF-α in jejunum tissue was examined. As shown in Fig. 3f, the intestinal expressions of F4/80 and TNF-α were slightly increased in CO-treated group after alcohol exposure, but remarkably increased in OFO-treated mice. OFO alone group showed the higher expressions, compared to CO alone mice, even CO plus ethanol group (Fig. 3f).

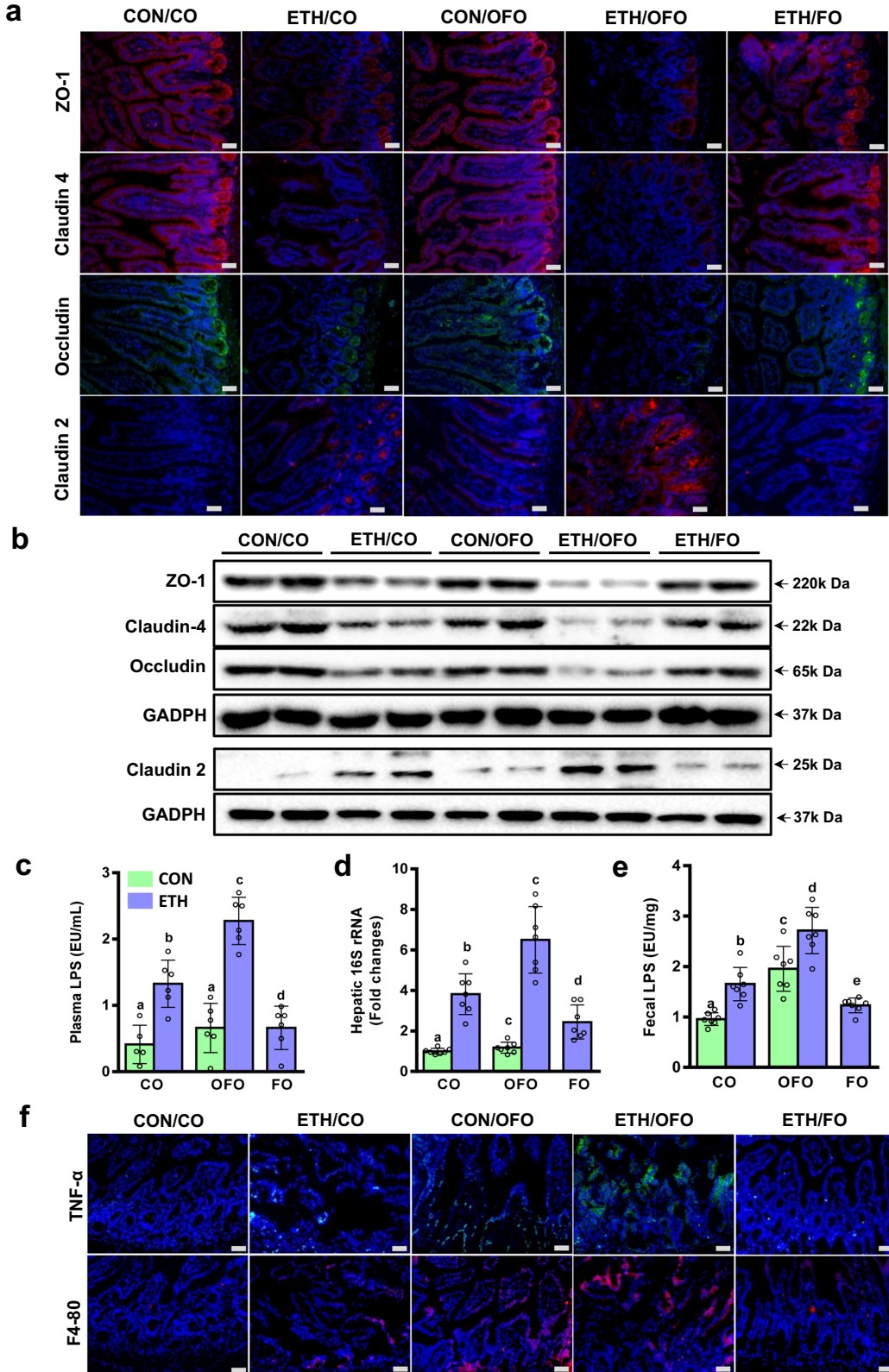

**Fig. 3 OFO enhances ethanol-induced intestinal barrier dysfunction. a** Immunofluorescent staining of intestinal TJ proteins, including ZO-1, claudin-4, occludin, and claudin-2, in jejunum tissue (scale bar, 25 μm). **b** Immunoblot analysis of intestinal TJ proteins. **c** Plasma LPS levels ($n = 5$–6). **d** The hepatic expressions of gram-negative bacteria 16S rRNA ($n = 7$). **e** Fecal LPS contents ($n = 7$). **f** Immunofluorescent staining of TNF-α (green) and F4/80 (red) in the sections of jejunum tissue (scale bar, 25 μm). Data were expressed as mean ± SD. Labeled means without a common letter differ within the column ($p < 0.05$). CO corn oil, OFO oxidized fish oil, FO fish oil.

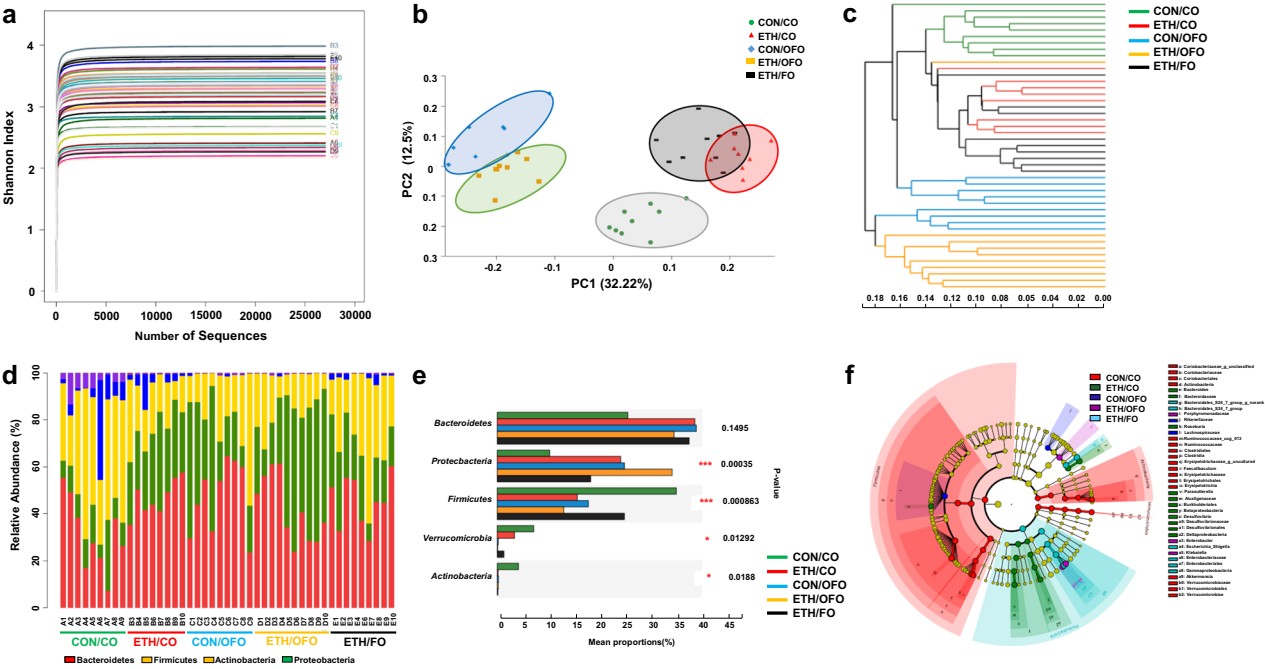

**Fig. 4 Bacterial 16S rRNA-based intestinal microbiome analysis. a** Rank abundance curve. **b** Principal coordinate analysis (PCoA) ($n = 6$–9). **c** Unweighted pair group method with arithmetic mean (UPGMA) analysis for the shifts in microbial community structure ($n = 8$–9). **d** Bar chart showing the relative abundance of intestinal microbiota on phylum level. **e** One-way ANOVA analysis of the composition of intestinal microbiota on phylum level ($n = 8$–9). **f** Cladogram analysis showed that taxa most differently associated with mice fed with CON or ETH diet (Wilcoxon rank-sum test). Circle sizes in the cladogram plot are proportional to bacterial abundance. The circle represents from the inner circle to the outer circle: phyla, genus, class, order, and family. CO corn oil, OFO oxidized fish oil, FO fish oil.

**OFO deteriorates alcohol-induced intestinal dysbiosis**. Intestinal dysbiosis and bacterial products play the crucial role in pathogenesis of ALD[5]. To whether intestinal dysbiosis contributes to OFO-aggravated alcoholic liver injury, intestinal microbiome was profiled by the analysis of bacterial 16S rRNA V3-V4 region. Rank abundance curve showed the sequencing depth was enough sufficient to characterize the bacterial communities in the samples (Fig. 4a). Principal coordinate analysis (PCoA) with unweighted UniFrac distance showed the tight clusters of the samples from each group. OFO-treated mice clustered closely regardless ethanol and pair feeding, which were far apart from other groups (Fig. 4b). Furthermore, ethanol-fed group, including ETH/CO and ETH/FO mice but except ETH/OFO, had similar microbiota profiles, and clustered separately with CON/CO. These results were confirmed by unweighted pair group method with arithmetic mean (UPGMA) tree (Fig. 4c), which indicated that the diversity of intestinal microbiota was slightly changed by ethanol feeding, but strongly influenced by dietary OFO. We next identified a specific intestinal microbiota related to OFO treatment. As shown in Fig. 4d, e, Phyla analysis showed that *Bacteroidetes*, *Proteobacteria*, and *Firmicutes* are dominant microbiota in all groups of mice, both ETH and OFO-fed mice had the increased *Bacteroidetes* and *Proteobacteria*, and the decreased *Firmicutes*, compared to CON/CO group. Furthermore, the proportion of *Proteobacteria* was slightly higher in CON/OFO mice (22%) over CON/CO mice (9.1%), and significantly increased in ETH/OFO mice (34%), while, FO treatment exhibited the moderate decrease in *Proteobacteria* (16%). Cladogram analysis showed that the OFO- and ETH-fed mice mainly clustered in *Proteobacteria* phylum, but exhibited at different taxonomic levels. The *beta-proteobacteria* and *delta-proteobacteria* family of *Proteobacteria* phylum was dominant in ETH/CO mice, while *grama-proteobacteria* family was dominant in OFO-fed mice.

Moreover, *Enterobacter* and *Klebsiella* genus was specifically clustered in ETH/OFO mice, but bacteria *Desulfovibrio* and *Parasutterela* were dominated in ETH/CO mice (Fig. 4f). These data indicated that OFO-exacerbated alcoholic liver injury might be associated with intestinal dysbiosis, especially the specific microbiota profile of *Proteobacteria*.

**ABx abolishes OFO-aggravated alcoholic liver injury**. To reveal the role of gut bacteria in OFO-aggravated liver injury in ethanol-fed mice, nonabsorbable antibiotics (ABx), comprising of polymyxin B and neomycin, were orally administered to the mice daily during liquid-diet adaptation and subsequent alcohol-feeding period. Nonabsorbable antibiotics did not alter the intestinal absorption and metabolism of ethanol, as measured by plasma ethanol level and hepatic expression of CYPE21 (Supplementary Fig. 5a, b). Gut sterilization by ABx significantly decreased LPS levels in feces and plasma, and bacterial translocation in both ETH/CO and ETH/OFO groups, but there is no difference between ETH/CO and ETH/OFO after ABx treatment (Fig. 5a–c). Immunofluorescent (Fig. 5d) and western blot (Fig. 5e) analysis indicated that the expressions of TJ proteins, ZO-1, occludin, claudin-4, and claudin-2 in jejunum tissue were reversed by ABx treatment in ethanol-fed mice, and OFO-aggravated intestinal barrier dysfunction was completely abolished by ABx treatment. Similar tendency was observed in immunofluorescent staining of F4/80 and TNF-α in jejunum tissue (Supplementary Fig. 6a).

Consistently, nonabsorbable antibiotics remarkably inhibited the hepatic inflammation in ethanol-fed mice, as evidenced by the reduced hepatic inflammatory cytokines TNF-α, IL-6, IL-1β (Fig. 6a), and MCP-1 (Fig. 6b), the less F4/80-positive monocytes/macrophage (Fig. 6c) in the liver, and the decreased hepatic

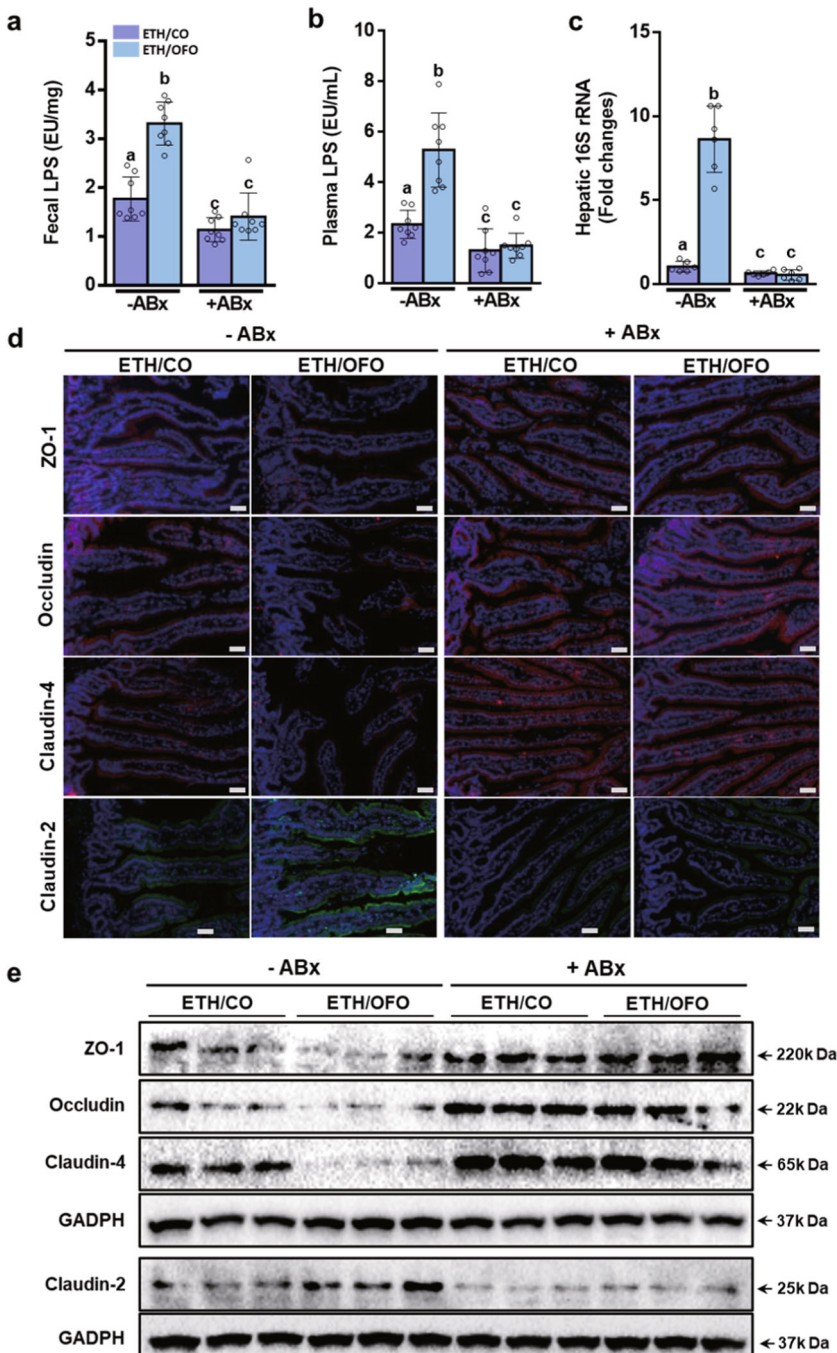

**Fig. 5 ABx abolishes OFO-aggravated intestinal barrier dysfunction in ethanol-fed mice.** a Fecal LPS contents ($n = 8$). b Plasma LPS levels ($n = 8$).
c Hepatic expression of 16S rRNA ($n = 6$). d Immunofluorescent staining. e Immunoblot analysis of intestinal TJ proteins, including ZO-1, occludin, claudin-4, and claudin-2 in the sections of jejunum tissue (scale bar, 25 μm). Data were expressed as mean ± SD. Labeled means without a common letter differ within the column ($p < 0.05$). CO corn oil, OFO oxidized fish oil, FO fish oil.

expressions of TLR4, MyD88, and p-p65 (Fig. 6d and Supplementary Fig. 6b). These results demonstrated that OFO-exacerbated hepatic inflammatory response was blunted by ABx treatment. Undoubtedly, intestinal sterilization with nonabsorbable antibiotics prevented hepatic steatosis and injury in alcohol-fed mice, as determined by plasma biochemical analysis (Fig. 7a), hepatic TG content assay (Fig. 7b), and histological staining of H&E and oil red O (Fig. 7c). These data suggest that the worse effects of OFO on alcoholic liver injury are resorted after ABx treatment.

**OCA blunts OFO-aggravated liver injury in alcohol-fed mice.** Alcohol and its metabolites disrupt intestinal barrier integrity, resulting in the increased permeability of intestinal barrier. Our data also indicated that OFO-induced intestinal inflammation may be involved in intestinal barrier dysfunction. Obeticholic acid (OCA), a potent agonist of farnesoid X receptor (FXR), is effective in the treatment of nonalcoholic steatohepatitis with liver fibrosis in animal models[34] and in patients[35]. OCA has been also shown to ameliorate gut barrier dysfunction and bacterial translocation in cholestatic[36] and in cirrhotic rats[37]. To further

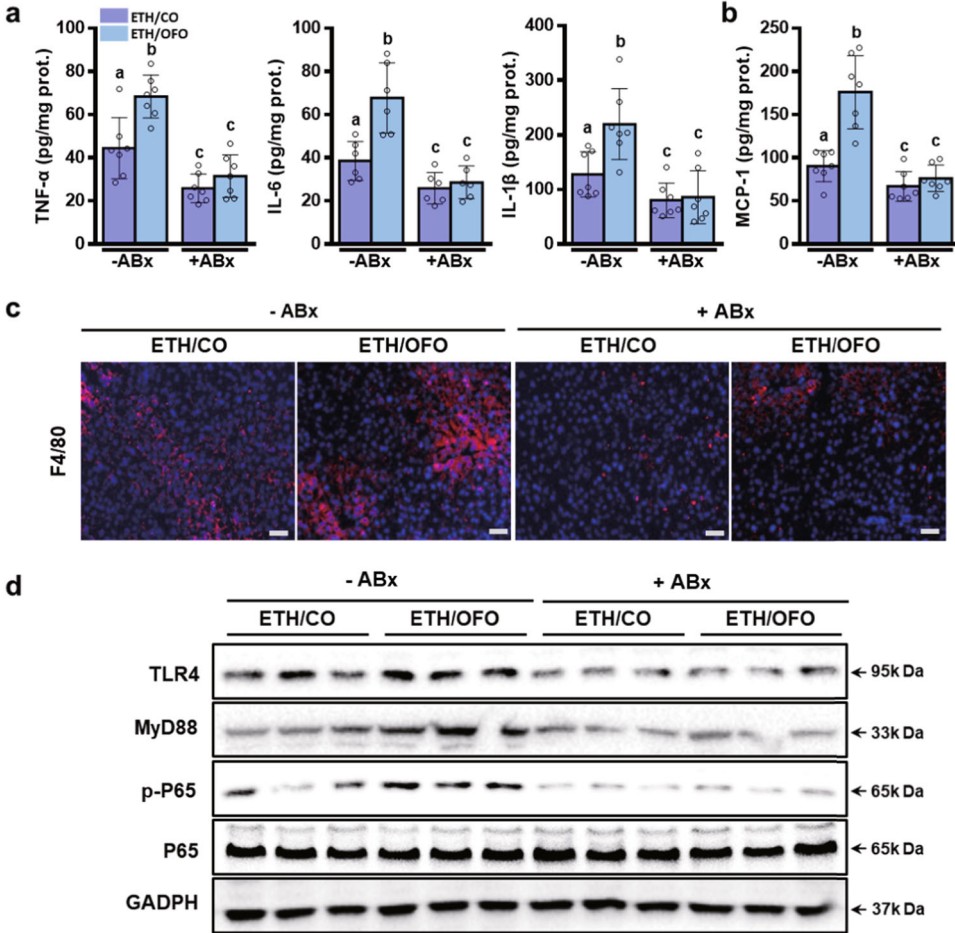

**Fig. 6 ABx treatment reverses OFO-aggravated hepatic inflammation in ethanol-fed mice. a** Hepatic level of cytokines (TNF-α, IL-6, IL-1β, and MCP-1) ($n = 6$–7). **b** Chemokine (MCP-1) ($n = 7$). **c** Immunofluorescent staining of F4/80 in the liver section (scale bar, 25 μm). **d** Hepatic expressions of TLR4, MyD88, p65, and phosphorylated p65 proteins. Value were expressed as mean ± SD. Labeled means without a common letter differ within the column ($p < 0.05$). CO corn oil, OFO oxidized fish oil, FO fish oil.

elucidate the role of intestinal barrier in OFO-aggravated alcoholic liver injury, OCA was gavaged daily (30 mg kg$^{-1}$) to the mice during alcohol-feeding period. As expected, FXR agonist OCA upregulated the expressions of TJ proteins, ZO-1, occludin and claudin-4 in jejunum tissue, and suppressed the intestinal inflammation in ethanol-fed mice (Supplementary Fig. 7). OCA treatment remarkably decreased LPS levels in plasma, but not changed in feces (Fig. 8a). Consistently, OCA inhibited the hepatic inflammation in ethanol-fed mice, as manifested by the reduced hepatic inflammatory cytokines TNF-α and IL-6 (Fig. 8b), and less F4/80-positive monocytes/macrophage in the section of liver tissue (Fig. 8c). Eventually, OCA treatment prevented hepatic steatosis and injury in alcohol-fed mice, as examined by histological analysis (Fig. 8d), plasma levels of AST and ALT (Fig. 8e), and hepatic TG quantification (Fig. 8f). These results demonstrate that stabilization of intestinal barrier by the treatment of FXR agonist OCA blunts OFO-aggravated liver injury in alcohol-fed mice.

## Discussion

In recent decades, an increasing number of studies has demonstrated that n-3 PUFAs exerted the beneficial effects on hepatic steatosis and liver injury induced by excess alcohol exposure[9,22–24]. However, the impact of n-3 PUFAs in ALD have been still inconsistent or controversial. Numerous studies indicated that the combination of n-3-rich FO and ethanol exacerbates hepatic

steatosis, inflammation, and liver injury, which was even used as pathological animal model for ALD[25–28]. The explanatory hypotheses proposed in these studies are that n-3 PUFAs in FO offer the preferential substrates for lipid peroxidation, further increasing oxidative stress and inflammation in the liver, eventually resulting in the exacerbated ALD[26–28]. While n-3 PUFAs with high degree of unsaturation, especially long-chain n-3 PUFAs, are more vulnerable to oxidation. The exogenous supplement of n-3 PUFAs might competitively reduce the potential of PUFAs in phospholipid bilayer of cell membrane to react with reactive oxygen species induced by alcohol exposure, leading to the protective effects against cell membrane damage. Interestingly, the positive results were invariably observed in the mouse-fed n-3 PUFAs with high purity[29], and fat-1 transgenic mice[22,38,39] in which n-3 PUFAs endogenously convert from n-6 PUFAs, thus avoiding PUFA oxidation ex vivo. It is unclear whether the exacerbated alcoholic liver injury observed in those studies with negative results is caused by n-3 PUFAs or its oxidative products already presenting in FO. Actually, the n-3 PUFA oxidation in the animal feeding stuffs was rarely concerned in the most of n-3 PUFA-related studies, likely contributing to the inconsistent outcomes[2]. In the present study, the probable impacts of the OFO on the pathogenesis of alcohol-induced liver injury were investigated. Our results demonstrated that alcohol-induced hepatic steatosis and inflammation are exacerbated by dietary OFO but prevented by unoxidized FO. The OFO-aggravated liver injury was associated with accentuating

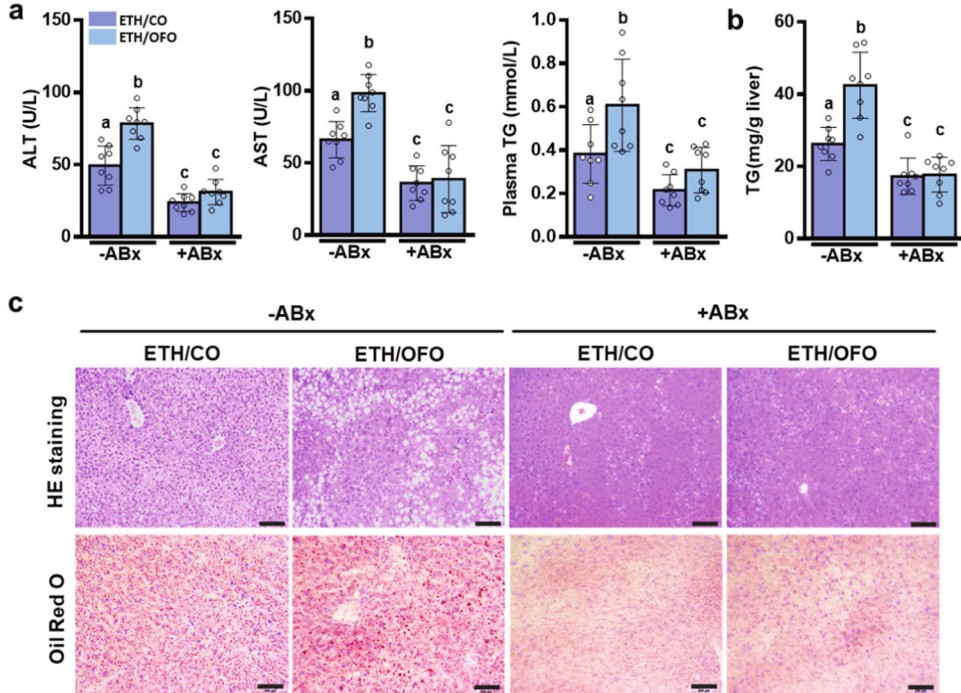

**Fig. 7 ABx treatment normalizes OFO-aggravated liver injury in ethanol-fed mice. a** Plasma levels of ALT, AST, and TG. **b** Hepatic TG contents.
**c** Representative H&E and Red Oil O staining of liver tissues (scale bar, 200 μm). Value was expressed as mean ± SD ($n = 8$). Labeled means without a common letter differ within the column ($p < 0.05$). CO corn oil, OFO oxidized fish oil, FO fish oil.

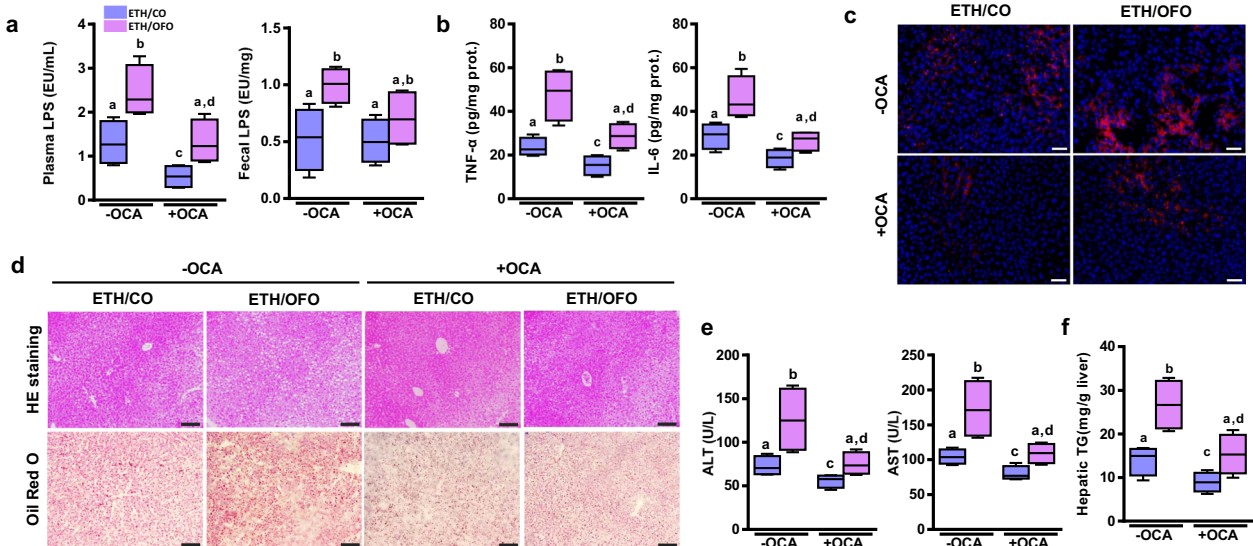

**Fig. 8 OCA blunts OFO-aggravated hepatic inflammation and liver injury in ethanol-fed mice. a** Fecal LPS contents. **b** Plasma LPS levels. **c** Inflammatory cytokines, TNF-α, and IL-6 in the liver. **c** Immunofluorescent analysis of hepatic F4/80 expression. **d** Representative H&E (scale bar, 100 μm) and Red Oil O staining of liver tissues (scale bar, 50 μm). **e** Plasma levels of ALT and AST. **f** Hepatic TG contents. Value was expressed as mean ± SD ($n = 4$). Labeled means without a common letter differ within the column ($p < 0.05$). CO corn oil, OFO oxidized fish oil, FO fish oil.

alcohol-induced intestinal dysbiosis and barrier dysfunction, subsequently enhancing gut-derived endotoxin-mediated inflammation in the liver.

Accumulating studies has demonstrated that gut microbiota dysbiosis plays a crucial role in the pathogenesis of ALD[5]. Alcohol exposure causes the overgrowth of pathogenic bacteria in the gut, in particular Gram-negative bacteria, leading to the overproduction of endotoxin[4]. It is well documented that gut microbiota manipulation, including fecal microbial transplantation and the treatment of probiotics or antibiotics, ameliorate endotoxemia, and liver injury induced by alcohol exposure in humans or rodent models[4,5]. In this study, sequencing of 1S ribosomal RNA indicated that both alcohol exposure and OFO induced a significant increases in the proportions of Gram-negative bacteria *Bacteroidetes* and *Proteobacteria*, and the decrease in *Firmicutes*, a Gram-positive bacteria, compared to the

pair-fed group, which are consistent with the previous studies[40,41]. Furthermore, the proportion of *Proteobacteria* was dramatically increased in ETH/OFO mice, and decreased in ETH/FO group, over alcohol-alone and OFO alone groups, indicating the alternation of *Proteobacteria* might be related to OFO treatment. Importantly, the specific bacteria taxa of *Proteobacteria* were discriminated between alcohol-fed mice and OFO-fed mice. These results were in line with the previous studies in which a high abundance of *Proteobacteria* was observed in fecal microbial communities of alcoholics and cirrhotic patients, and the degree of its overgrowth was correlated with the severity of the liver injury[42]. *Proteobacteria* has been considered as the microbial signature of intestinal dysbiosis, and its composition alternation was also observed in patients with intestinal inflammation-related diseases and metabolic syndrome[43]. However, the potential reason for the bacterial overgrowth caused by OFO remains to be further investigated. In addition, PCoA analysis showed that ETH/CO and ETH/FO group exhibited close microbiota profiles and clustered separately with CON/CO (Fig. 4b), which was confirmed by UPGMA analysis (Fig. 4c). These results indicated that overall composition of intestinal microbiota was greatly disturbed by alcohol exposure regardless CO- and FO-feeding, and FO treatment failed to recover microbial community structure influenced by alcohol. However, phyla analysis showed that FO treatment exhibited the significant decrease in the specific bacteria taxa of *Proteobacteria* that greatly elevated by alcohol exposure. Collectively, FO treatment may inhibit alcohol-induced overgrowth of *Proteobacteria*, but not largely change microbial community structure, which contribute to the protective effects of FO against alcoholic liver injury.

The overgrowth of gut microbiota, especially Gram-negative, increases the bacterial products, endotoxin, or LPS. A large body of clinical and animal studies has supported that the circulating levels of gut-derived LPS is closely correlated with the development of liver injury induced by alcohol exposure[7,8]. Gut-derived LPS diffuses via the disrupted intestinal barrier from the intestine into the portal vein, subsequently activates hepatic Kupffer cells by TLR4 signaling pathway, resulting in inflammation in the liver[9]. In this study, alcohol exposure caused the increases in the LPS levels in both plasma and feces, these increases were further enhanced by dietary OFO. Consistently, OFO aggravated alcohol-induced hepatic inflammation and liver injury, these effects were completely abolished by intestinal sterilization using nonabsorbable antibiotics. Our findings suggest that OFO-exacerbated ALD is associated with intestinal dysbiosis and gut-derived LPS-mediated hepatic inflammation.

Another contributor to alcoholic endotoxemia is gut leakiness. Alcohol and its metabolite acetaldehyde disrupt the integrity of the intestinal epithelial cell layer and increase intestinal permeability, facilitating the diffusion of endotoxin and translocation of intestinal bacteria. Besides the elevated plasma LPS level, we observed the increased hepatic expression of bacteria 16S rRNA, a signature of bacterial translocation, in alcohol-fed mice. It is intriguing that OFO alone group has the comparable plasma LPS, but significantly increased level of fecal LPS, indicating that the high level of intestinal LPS induced by OFO is not allowed to influx into the portal vein. A most plausible explanation is that OFO alone could not impair intestinal integrity, which was further confirmed by immunofluorescent and western blot analysis of intestinal TJ proteins. These results suggest that the main functions of OFO and alcohol are to induce intestinal dysbiosis and intestinal barrier dysfunction in OFO-aggravated alcoholic liver injury, respectively.

TJ proteins, including ZO-1, occludins, and claudins, serve as specialized junctional complexes that offer physical intestinal barrier. Acetaldehyde induces tyrosine phosphorylation of TJ proteins, leading to intestinal hyperpermeability to macromolecules. The downregulated intestinal TJ proteins were observed in rodent models and the patients with ALD[44]. In this study, alcohol exposure impaired intestinal barrier function, as evidenced by the deceased expression of ZO-1, occludin, and claudin-4, and the increased expression of claudin-2, a mediator of barrier dysfunction during intestinal inflammation[45], which was further aggravated by combination of OFO and alcohol. However, OFO alone did not affect the TJ protein expression, suggesting that beside the action of acetaldehyde, another factor might be implicated in intestinal barrier dysfunction. Accumulating clinical and experimental evidence has indicated that intestinal inflammation induced by dysbiosis and TNF-α receptor I signaling in epithelial cells contributes to disruption of the intestinal barrier[46,47]. Our data indicated that dietary OFO significantly increased intestinal expression of TNF-α and F4/80, suggesting that intestinal inflammation is implicated in OFO-deteriorated intestinal barrier dysfunction in alcohol-fed mice.

Collectively, our data clearly show that the OFO exacerbates alcoholic liver injury via enhancing intestinal dysbiosis, barrier dysfunction, and hepatic inflammation mediated by gut-derived LPS in mice, as illustrated in Fig. 9. To our knowledge, this study provides the new insight into the inconsistent outcome regarding the role of n-3 PUFAs in ALD and other diseases.

## Methods

**OFO preparation**. FO (45.5% EPA, 33.5% DHA, and 13% other n-3 PUFAs) was purchased from KangDao Biotechnology Limited Company (Shenzheng, China). OFO was prepared by heating the fresh FO on a watch glass at 65 °C for 2 weeks for the adequate oxidation.

**Measurements of lipid oxidation and fatty acid profile**. POV assay was performed by visual titration of iodine method to evaluate the primary lipid-oxidation products as described previously[48]. The POV was expressed as milliequivalents peroxide per kilogram (meq kg$^{-1}$). Measurement of p-anisidine value (AV) was determined using a colorimetric method[49]. The results of AV were calculated according to the formula "AV = $10 \times [1.2 \times (A2 - A1)]/SW$, where SW is the sample weight (g). The POV and AV were obtained to evaluate the total oxidation value (Totox) according to the formula "Totox = POV × 2 + AV." In order to measure the second oxidation products in the oil samples, the TBARS assay was conducted with minor modifications as described previously[50]. The malonaldehyde (MDA) was used as the standard. TBARS were expressed as mg MDA equivalents/kg sample. In addition, the fatty acid profiles in FO and OFO were measured by GC-MS, as previously described[51].

**Animal model and drug administration**. Male C57BL/6 mice (10–12 weeks old) were housed in specific pathogen-free room at Experimental Animal Center, Guangdong Pharmaceutical University (Guangzhou, China). Alcoholic liver injury was induced by a NIAAA model of chronic-plus-binge alcohol feeding as described previously[52]. Briefly, after 1 week of acclimatization with liquid control diet, all mice were fed the modified Lieber-DeCarli liquid diets containing ethanol (ETH) or isocaloric maltose dextrin as the CON diet (TROPHIC Animal Feed High-Tech Co., Ltd Nantong, Jiangsu, China), supplementing with CO or OFO for 10 days. CO was used as fat source in the modified Lieber-DeCarli liquid control diets to eliminate the possible effects of different energy intake and other type of fats. An additional group of mice received ethanol liquid-diet containing unoxidized FO was used to compare the effects of OFO. Diet compositions were shown in Supplementary Table 2. On day 11, ethanol-fed mice and control mice were received with a single dose of ethanol (31.5%, *v/v*, 5 g kg$^{-1}$) and an isocaloric maltose dextrin, respectively. After 9-h fasting, all mice were euthanized, and plasma, liver, intestine, and fresh feces were harvested for further analysis. Animal feeding process was illustrated in Supplementary Fig. 2a. Animal practices were approved by the Animal Ethics Committee, Institute of Chinese Medical Sciences, University of Macau.

The second group of C57BL/6 male mice was fed the Lieber-DeCarli alcohol liquid diets supplemented with CO or OFO, nonabsorbable antibiotics (ABx; Polymyxin B, 150 mg kg$^{-1}$ BW and Neomycin, 200 mg kg$^{-1}$ BW) or an equal volume of vehicle were orally administered to the mice daily during liquid-diet adaptation and subsequent alcohol-feeding period (Supplementary Fig. 2b)[16].

The third group of C57BL/6 male mice was fed the Lieber-DeCarli alcohol liquid diets supplemented with CO or OFO, the mice were gavaged with OCA (30 mg kg$^{-1}$) or equal volume of vehicle during the period of alcohol feeding (Supplementary Fig. 2c).

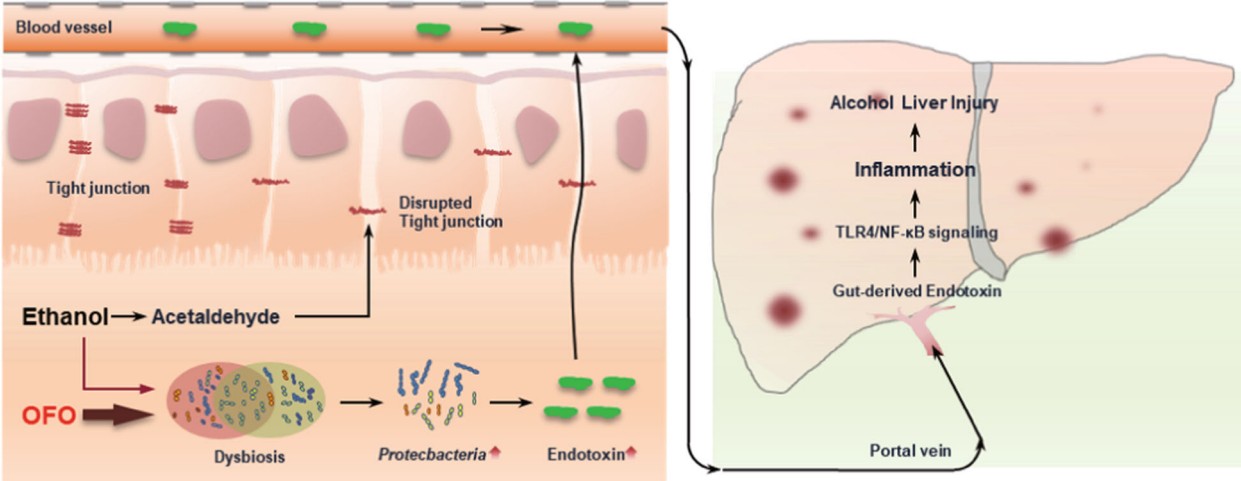

**Fig. 9 Schematic diagram of the potential mechanisms underlying OFO-aggravated alcoholic liver injury.** Alcohol and its metabolite, acetaldehyde, mainly disrupt intestinal integrity by down-regulation of intestinal epithelial tight junction proteins. OFO exacerbates intestinal dysbiosis, particularly the overgrowth of bacteria Proteobacteria, leading to the increased bacterial endotoxin. The impaired intestinal barrier facilitates endotoxin translocation from the intestinal lumen into the portal circulation. Subsequently, the gut-derived endotoxin activates hepatic Kupffer cells by TLR4-mediated NF-κB signaling pathway, resulting in hepatic inflammation, which induces hepatocyte damage. In summary, OFO exacerbates alcoholic liver injury via enhancing intestinal dysbiosis, barrier dysfunction and hepatic inflammation mediated by gut-derived endotoxin in mice. NF-κB nuclear factor κB, TLR4 Toll-like receptor 4, OFO Oxidized fish oil.

**Biochemical and histological analysis**. Plasma activities of ALT and AST were determined using the corresponding commercial reagent kits (Nanjing Jianchen Bioengineering Institute, Nanjing, China). TG level and ethanol concentration in plasma were analyzed by commercially available quantification kits (BHKT Clinical Reagent Co., Ltd, Beijing, China) and an Ethanol Assay Kit following the manufacturers' instructions (BioVision, Milpitas, CA, USA), respectively. LPS levels in plasma and feces were determined using a commercially available ELISA kit (Cloud-Clone Corp., Business Co., Ltd, Wuhan, China). Histopathological changes of liver were observed by H&E staining as standard protocol elsewhere.

**Measurement of hepatic steatosis**. To evaluate hepatic lipid accumulation, liver samples were determined by Oil Red O staining and triacylglycerol quantification. After fixing in 4% phosphate-buffered paraformaldehyde, frozen sections (8 μm) were stained with Oil Red O (Sigma-Aldrich, St Louis, USA) for 10 min and subsequently counterstained with hematoxylin according to a standard protocol. For the quantification of hepatic TG level, 50 mg of liver sample was homogenized and mixed with chloroform/methanol (2:1, $v/v$) overnight. After centrifugation, the low lipid phase was dissolved in chloroform, and was collected to determine lipid level using the TG quantification kit (Beijing BHKT clinical reagent Co., Ltd, Beijing, China). Hepatic TG content was normalized to correspondingly liver weight and expressed as mg kg$^{-1}$.

**Measurements of oxidative stress parameters and cytokines**. Liver homogenates were prepared in the cold radioimmunoprecipitation assay (RIPA) buffer (Beyotime Institute of Biotechnology, Nanjing, China) following the manufacturer's instructions. The enzymatic activities of CAT and SOD, and levels of hepatic GSH and MDA by commercially available kits according to the manufacturer's instruction (Nanjing Jiancheng Bioengineering Institute, Nanjing, China). For measurement of the hepatic inflammatory cytokines and chemokines, the liver tissues were homogenized and collected to determine the levels of TNF-α, IL-1β, IL-10, and MCP-1 using Mouse ELISA MAX™ Standard kits according to the manufacturer's instruction (BioLegend Inc., San Diego, CA, USA), respectively. All values were normalized to hepatic total protein.

**Immunofluorescent staining**. Immunofluorescence analysis for the expression of F4/80 and p65 in liver tissue, and TJ proteins in jejunum tissue were performed as described[9]. In brief, the cryosections of liver or intestine tissues (8 μm) were fixed in cold acetone for 15 min, washed with phosphate-buffered saline (PBS) for three times, and blocked nonspecific binding with 5% normal goat serum in PBS. For p65 staining, permeabilization is prior to block the endogenous peroxidase. After incubation with primary antibodies (1:100) overnight at 4 °C, the sections were washed with PBS three times, subsequently incubated with Alexa Fluor® 568 or Alexa Fluor® 488 goat anti-rabbit IgG (1:200, Life Technologies) at room temperature for 2 h in dark. Nuclei were stained with DAPI, followed by visualization using a Leica TCS SP8 confocal laser scanning microscope (Leica Microsystems, Wetzlar, Germany).

**Immunoblot assay**. Total proteins of partial liver and jejunum tissues were extracted with RIPA lysis buffer. Equivalent amount of protein samples in each group were separated on 8–10% sodium dodecyl sulfate-polyacrylamide gel electrophoresis. After electrophoresis, proteins were transferred into polyvinylidene difluoride membrane (Millipore, Bedford, MA, USA). After blocking with 5% non-fat dry milk, the membranes were incubated with specific primary antibodies (Table S3) independently overnight at 4 °C, and then incubated with corresponding horseradish-peroxidase-conjugated secondary antibodies.

**Quantitative PCR analysis**. Total RNA was extracted from homogenized liver samples using commercial Trizol reagent (Takara Bio., Tokyo, Japan) according to the manufacturer's protocol. Quantitative PCR were conducted using SYBR green (Applied Biosystems) in a Stratagene Mx3005P multiple quantitative PCR system (Agilent Technologies, Santa Clara, CA, USA). The sequences for the primer pair of 16S rRNA were shown as follows (Forward: 5′-AGAGTTTGATCCTGGC TC AG-3′, Reverse: 5′-TGCTGCCTCCCGTAGGAGT-3′), and 18S rRNA (Forward: 5′-ACGG ACCAGAGCGAAAGCAT-3′, Reverse: 5′-TGTCAATCCTGTCCGTG TCC-3′). The gene expression level of 16S rRNA was normalized to 18S *RNA*.

**Microbial analysis**. Fresh fecal pellets from individual mice were collected from each mice immediately upon defecation, frozen subsequently stored at −80 °C. Bacterial genomic DNA was extracted from fecal samples using the Power Fecal™ DNA Isolation kit (MO BIO Laboratories, Carlsbad, CA). DNA samples were sequenced and amplified using the hypervariable regions of V3-V4 of bacterial 16S ribosomal DNA (PCR1F_460: 5′-CTTTCCCTACACGACGCTCTTCCGATCTA CGG RAGGCAGCAG-3′, PCR1R_460: 5′-GGAGTTCAGACGTGTGCTCTTCC GATCTTACCAGGGT ATCTAATCCT-3′). The amplicons of PCR products were extracted on agarose gels and purified using an Axygene DNA gel extraction kit (Coring, NY, USA). The purified amplicons were sequenced on Illumina Miseq 2500 sequencing platform (Major BioPharm Technology, Shanghai, China). The raw sequences from original DNA fragments were assembled using Trimmomatic and FLASH programs, and then filtered the default parameters using the quantitative insights into microbial ecology (QIIME v1.9.0)[31]. The non-chimeric sequences were demultiplexed clustered into species-level (97% similarity) operational taxonomic units (OTUs) using a de novo OTU picking protocol. Representative sequence for corresponding OTU was identified to annotate taxonomic assignment using the RDP classifier (version 2.2)[53]. Analysis of Simpson's diversity index, Shannon–Wiener index, and rarefaction estimates were calculated using QIIME. Bacterial alpha diversity was performed using Shannon Index. Analysis of the prevalence of OTUs <5% were discarded. Data process using R software v2.14.1 restricted the same taxonomic assignment in the merged OTUs. Results are represented as the mean ± SEM. Beta diversity was performed using unweighted unifrac distances visualizing by PCoA and UPGMA. Cladograms generated from linear discriminant analysis coupled with effect size were performed to identify the most differentially abundant taxa between groups at the different taxonomy levels.

**Statistics and reproducibility**. Statistical analyses were conducted using Graph-Pad Prism 6.0 package (San Diego, CA, USA). All data, except in microbial analysis, are presented as mean ± standard deviation. One-way analysis of variance with the Student's $t$ test or Tukey's post hoc test was performed to evaluate the difference between groups. The value of $p < 0.05$ was considered statistically significant. All experiments, except microbial analysis, were conducted in at least three independent replicates. The sample size is indicated for each experiment in the corresponding figure legend.

**Reporting summary**. Further information on research design is available in the Nature Research Reporting Summary linked to this article.

## Data availability

Data supporting the findings of this work are available within the paper and its Supplementary files or available from the corresponding author upon reasonable request. Raw data underlying plots in figures are available in Supplementary Data 1. The unprocessed blot images with size markers are provided in Supplementary Information.

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

## Acknowledgements

This work was financially supported by the grants from the Research Committee of the University of Macau (MYRG2017-00035-ICMS) and the Science and Technology Development Fund, Macau SAR (File no. 065/2018/A2 and 0034/2019/A1).

## Author contributions

J.-B.W. conceived and designed the experiments, and finalized the manuscript; R.F., L.M., M.W., C.L., and R.Y. performed experiments and acquired data; R.F. wrote the initial manuscript; and H.S. and Y.Y. contributed to statistical analysis. All authors have read, commented on, and approved the final paper.

## Competing interests

The authors declare no competing interests.
