## [Peer Review File · Communications Biology]

Reviewers' comments:

Reviewer #1 (Remarks to the Author):

The study demonstrated that the oxidized fish oil (OFO) exacerbates hepatic steatosis and liver injury in mouse model of acute-on-chronic alcohol feeding, which were associated with the enhanced intestinal dysbiosis, barrier dysfunction, and hepatic inflammation mediated by gut-derived endotoxin. It is well known that n-3 PUFA exert the beneficial effects in many conditions or diseases, but the role of n-3 PUFAs in alcoholic liver disease (ALD) has been still controversial. A large number of studies has demonstrated that n-3-rich fish oil (FO) promotes the pathogenesis of ALD, which is attributed to the intensified oxidative stress and inflammation in the liver. But n-3 PUFA is highly susceptible to oxidation and exert anti-inflammatory activity. It is unclear whether the high levels of oxidative stress and inflammation observed in these studies is caused by n-3 PUFAs or its oxidation products already existing in fish oil used. Thus, the rationale of this study seems to be sound. It is a meaningful work providing the insight into the inconsistent outcome regarding the role of n-3 PUFA in ALD and other diseases. Some comments are offered for author's consideration to improve the manuscript.

1. The use of corn oil (CO) as the control for FO and OFO should be justified.
2. OFO was prepared by air exposure at 65 °C for two weeks. It could not well simulate the oxidation degree in the fish oil capsule.
3. Except the parameter of oxidation degree, POV, AV, Totox, and TBARS, the fatty acid profiles of fresh FO and OFO should be provided.
4. The observation on OFO effects in alcoholic liver injury is convincing, but it is not clear to which extent the observations would affect non-alcoholic injury (e.g. NAFLD).
5. Why did the OFO alone group shown the higher levels of ALT, AST and hepatic inflammatory cytokines, but without change of TG accumulation? Please address this issue.
6. As for the H&E staining results in Figure 1, all photos should be taken from the same area such as CV area or PV area.
7. The complete language check is required throughout the whole manuscript.

Reviewer #2 (Remarks to the Author):

In this manuscript entitled "Oxidized fish oil exacerbates alcoholic liver disease by enhancing intestinal dysbiosis and barrier dysfunction", Feng et al. present data indicating that oxidized fish oil worsens alcohol induced liver injury, while unoxidized fish oil is protective. The author's hypothesis surrounding the use of fish oil is bolstered by the interesting link to the cause of injury, the change of intestinal permeability and host microbiome. While parts of the hypothesis have been published by others, the authors present intriguing data that brings together the intricate data. That being said, this is an intriguing and novel paper. However, there are several MINOR concerns that need to be addressed.

*Further clarity of the statistical analysis is necessary to easily understand the graphs. The lettering system is confusing and difficult to follow. Several Figure legends are hard to follow especially when it comes to statistical analysis.

*In Figure 2A, the IL-10 was not rescued by FO while other cytokines in this panel were. While the data in the later figures support this finding, we believe an explanation in the text may be necessary.

*While the authors provide evidence for a leaky intestinal barrier, a standard experiment to study barrier integrity with FITC-dextran method would strengthen the data, but may not be necessary.

*Figure 4 shows similarities between ETH and ETH+FO group. This was surprising since the data

shows a complete rescue by FO for most parameters. Further discussion of this is necessary to understand this discrepancy.

*While OCA data is interesting, further explanation of the use of this model in this paper is necessary. The reason is OCA has been shown to have an effect on liver fibrosis. Further discussion is needed to understand the rationale of the use of this to support the hypothesis.

Reviewer # 1

Overall: “...the rationale of this study seems to be sound. It is a meaningful work providing the insight into the inconsistent outcome regarding the role of n-3 PUFA in ALD and other diseases. Some comments are offered for author’s consideration to improve the manuscript.”

- We greatly appreciate you for your careful review and valuable comments which greatly improved our manuscript.

1. *The use of corn oil (CO) as the control for FO and OFO should be justified.*

- Energy consumption is a crucial factor that greatly affect the experimental outcomes in metabolic disorder, including alcoholic liver injury. Due to be rich in n-6 PUFAs, corn oil (CO) has been widely used as a control for n-3 PUFAs in numerous animal studies to eliminate the possible effects of different energy intake and other type of fats (such as saturated fat or monounsaturated fats).
- The corresponding description has been included in the current manuscript.

2. *OFO was prepared by air exposure at 65 °C for two weeks. It could not well simulate the oxidation degree in the fish oil capsule.*

- We sincerely appreciate the valuable and professional comments.
- We totally agree with the reviewer’s opinion that OFO used in this study may not faithfully

mimic natural oxidation that occur in the fish oil capsule. In this study, oxidized fish oil was prepared as the method reported previously with modification [1].

- Considering that unoxidized fish oil may exert the protective effect against alcoholic liver injury, and counteract the OFO-exacerbated effects, the oxidation of fish oil should be sufficient. Although n-3 PUFAs are highly unstable and vulnerable to be oxidized more easily than other types of fatty acids, vitamin E was commonly added in fish oil capsule to protect n-3 PUFA against oxidation. Our data indicated that approximately 50.9 % EPA and 58.7 % DHA in FO were oxidized after air exposure at 65 °C for two weeks.
 - Although FO oxidation condition used in current study may not faithfully mimic natural oxidation that occur in fish oil capsule, we believe it is not critical for the strength of our conclusion, as our objective was to determine the impacts of **oxidized fish oil** on alcoholic liver injury.
3. *Except the parameter of oxidation degree, POV, AV, Totox, and TBARS, the fatty acid profiles of fresh FO and OFO should be provided.*
- As suggested, the fatty acid profiles of both fresh FO and OFO were determined by chromatography–mass spectrometry (GC-MS) as our previously method. The results were shown in **Table S1** and **Fig. S2** in the revised manuscript.
 - As determined by GC-MS, the contents of n-3 PUFAs, including EPA, DPA and DHA, in OFO were greatly decreased, compared to unoxidized FO. Approximately 50.9 % EPA and 58.7 % DHA in FO were oxidized during the process.
4. *The observation on OFO effects in alcoholic liver injury is convincing, but it is not clear to which extent the observations would affect non-alcoholic injury (e.g. NAFLD).*
- In the current study, oxidized fish oil exacerbates alcoholic liver disease by enhancing intestinal dysbiosis, barrier dysfunction, and hepatic inflammation mediated by gut-derived LPS in mice. Given the fact that these factors were also implicated in the pathogenesis of NAFLD as previously reported [3-4], we speculate that OFO exhibit the deteriorative effects on NAFLD. However, the specific impacts of OFO on NAFLD remains to be investigated in the future.
5. *Why did the OFO alone group shown the higher levels of ALT, AST and hepatic inflammatory cytokines, but without change of TG accumulation? Please address this issue.*
- We sincerely appreciate the valuable comments.
 - OFO alone treatment slightly caused the increased plasma levels ALT and AST, the circulating markers of liver injury, and hepatic inflammatory cytokines, suggesting that OFO

alone lead to less severe liver injury.

- Our data indicated that OFO-treated mice showed the increasing tendency in both hepatic and plasma TG contents, compared with CO-treated mice (8.96 ± 2.16 vs 6.84 ± 1.91 , $p=0.064$; 0.25 ± 0.087 vs 0.20 ± 0.067 , $p=0.32$), but there was no significant difference. These data indicate that hepatic steatosis play little role in OFO-caused liver injury.
6. *As for the H&E staining results in Figure 1, all photos should be taken from the same area such as CV area or PV area.*
 - As suggested, H&E staining figures have been replaced accordingly.
 7. *The complete language check is required throughout the whole manuscript.*
 - We have carefully reviewed the manuscript and corrected the grammatical and typing errors as much as possible.

Reviewer # 2

Overall: *In this manuscript entitled “Oxidized fish oil exacerbates alcoholic liver disease by enhancing intestinal dysbiosis and barrier dysfunction”, Feng et al. present data indicating that oxidized fish oil worsens alcohol induced liver injury, while unoxidized fish oil is protective. The author’s hypothesis surrounding the use of fish oil is bolstered by the interesting link to the cause of injury, the change of intestinal permeability and host microbiome. While parts of the hypothesis have been published by others, the authors present intriguing data that brings together the intricate data. That being said, this is an intriguing and novel paper. However, there are several MINOR concerns that need to be addressed.*

- We greatly appreciate the reviewer for constructive criticisms and valuable comments, which were of great help in revising the manuscript.
1. *Further clarity of the statistical analysis is necessary to easily understand the graphs. The lettering system is confusing and difficult to follow. Several Figure legends are hard to follow especially when it comes to statistical analysis.*
 - We sincerely appreciate the valuable and professional comments.

- Due to 4-5 groups in this study, we used the lettering system to mark significant differences between groups. Two groups without a common letter are statistically significant at $p < 0.05$. The lettering system is one of widely used methods for statistical analysis.
 - We also attempted to use different method to mark significant differences between groups. Eventually, we believe that the lettering system is the best way to indicate significant differences in a bar chart plot for the pairwise comparisons of 4-5 groups. We apologized that we cannot find the better and clearer approach.
2. *In Figure 2A, the IL-10 was not rescued by FO while other cytokines in this panel were. While the data in the later figures support this finding, we believe an explanation in the text may be necessary.*
- Our data indicated that FO supplement significantly decreased pro-inflammatory cytokines (TNF- α , IL-6 and IL-1 β) and chemokine (MCP-1) induced by ethanol exposure, but FO did not rescue hepatic level of IL-10, an anti-inflammatory cytokine released by Kupffer cells and monocytes, which is paralleled with previous study [2]. We speculated that the inhibition of pro-inflammatory cytokines, but not normalization of anti-inflammatory cytokines, contribute to the protective effects of FO against ALD in the mouse model of chronic-plus-single-binge ethanol feeding. IL-10 plays a critical role in the regulation of immune cells, especially regulatory T (Treg) cells, to keep maintain immune balance in ALD. The role of IL-10 in the protective effects of FO remains to be further investigated.
 - The explanation has been now included in the current manuscript.
3. *While the authors provide evidence for a leaky intestinal barrier, a standard experiment to study barrier integrity with FITC-dextran method would strengthen the data, but may not be necessary.*
- We sincerely appreciate the valuable and professional comments.
 - We totally agree with the reviewer's opinion that FITC-dextran method is a standard experiment to examine the intestinal barrier integrity and it would strengthen our findings. In this study, the alternative methods were performed to provide the evidence of intestinal barrier dysfunction, including the expressions of tight junction proteins, the circulating LPS and bacteria translocation (hepatic 16S rRNA). These measurements have also been widely used to evaluate the intestinal barrier integrity.
4. *Figure 4 shows similarities between ETH and ETH+FO group. This was surprising since the data shows a complete rescue by FO for most parameters. Further discussion of this is necessary to understand this discrepancy.*

- We sincerely appreciate the valuable and professional comments.
 - PCoA analysis showed that ETH/CO and ETH/FO group exhibited close microbiota profiles and clustered separately with CON/CO (**Fig. 4B**), which was confirmed by UPGMA analysis (**Fig. 4C**), indicating that **overall composition of the gut microbiome** was greatly disturbed by alcohol exposure regardless CO- and FO-feeding, and FO treatment failed to recover microbial community structure influenced by alcohol. However, further phyla analysis showed FO treatment exhibited the significant decrease in the specific bacteria taxa of *Proteobacteria* that greatly elevated by alcohol exposure. *Proteobacteria* has been considered as the microbial signature of intestinal dysbiosis, and its composition alternation were also observed in patients with intestinal inflammation-related diseases and metabolic syndrome [3]. Collectively, FO treatment may inhibit alcohol-induced overgrowth of *Proteobacteria*, but not largely change the overall composition of the gut microbiome, which contribute to the protective effects of FO against alcoholic liver injury.
 - Additionally, our data also indicated that FO supplement significantly reversed the expressions of intestinal TJ proteins, including ZO-1, claudin-4, and occludin, upregulated by alcohol exposure, which also contribute to its protective effects.
 - The discussion has been now included in the revised manuscript.
5. *While OCA data is interesting, further explanation of the use of this model in this paper is necessary. The reason is OCA has been shown to have an effect on liver fibrosis. Further discussion is needed to understand the rationale of the use of this to support the hypothesis.*
- In this study, we hypothesize that both intestinal dysbiosis and barrier dysfunction are associated with OFO-exacerbated liver injury. Thus, **ABx and OCA treatment experiments were designed to verify the roles of intestinal dysbiosis and barrier dysfunction in OFO-aggravated liver injury in alcohol-fed mice, respectively.**
 - Alcohol and its metabolites disrupt intestinal barrier integrity, resulting in the increased permeability of intestinal barrier. Our data also indicated that OFO-induced intestinal inflammation may be involved in intestinal barrier dysfunction. Obeticholic acid (OCA), a potent agonist of farnesoid X receptor (FXR), is effective in the treatment of nonalcoholic steatohepatitis (NASH) with liver fibrosis in animal models [4] and in patients [5]. OCA has been also shown to ameliorate gut barrier dysfunction and bacterial translocation in cholestatic [6] and in cirrhotic rats [7]. To further elucidate the role of intestinal barrier in OFO-aggravated alcoholic liver injury, OCA was gavaged daily (30 mg/kg) to the mice during alcohol feeding period. Our data demonstrated that stabilization of intestinal barrier by the treatment of OCA significantly blunt OFO-aggravated liver injury in alcohol-fed mice.

Reference

- [1] C. Song, B. Liu, P. Xu, J. Xie, X. Ge, Q. Zhou, C. Sun, H. Zhang, F. Shan, Z. Yang, Oxidized fish oil injury stress in *Megalobrama amblycephala*: Evaluated by growth, intestinal physiology, and transcriptome-based PI3K-Akt/NF-kappaB/TCR inflammatory signaling, *Fish Shellfish Immunol*, 81 (2018) 446-455.
- [2] X. Zhang, H. Wang, P. Yin, H. Fan, L. Sun, Y. Liu, Flaxseed oil ameliorates alcoholic liver disease via anti-inflammation and modulating gut microbiota in mice, *Lipids Health Dis*, 16 (2017) 44.
- [3] Y. Litvak, M.X. Byndloss, R.M. Tsohis, A.J. Bäumlér, Dysbiotic Proteobacteria expansion: a microbial signature of epithelial dysfunction, *Current opinion in microbiology*, 39 (2017) 1-6.
- [4] H. Jouihan, S. Will, S. Guionaud, M.L. Boland, S. Oldham, P. Ravn, A. Celeste, J.L. Trevaskis, Superior reductions in hepatic steatosis and fibrosis with co-administration of a glucagon-like peptide-1 receptor agonist and obeticholic acid in mice, *Mol Metab*, 6 (2017) 1360-1370.
- [5] S. Mudaliar, R.R. Henry, A.J. Sanyal, L. Morrow, H.U. Marschall, M. Kipnes, L. Adorini, C.I. Sciacca, P. Clopton, E. Castelloe, P. Dillon, M. Pruzanski, D. Shapiro, Efficacy and safety of the farnesoid X receptor agonist obeticholic acid in patients with type 2 diabetes and nonalcoholic fatty liver disease, *Gastroenterology*, 145 (2013) 574-582 e571.
- [6] L. Verbeke, R. Farre, B. Verbinen, K. Covens, T. Vanuytsel, J. Verhaegen, M. Komuta, T. Roskams, S. Chatterjee, P. Annaert, I. Vander Elst, P. Windmolders, J. Trebicka, F. Nevens, W. Laleman, The FXR agonist obeticholic acid prevents gut barrier dysfunction and bacterial translocation in cholestatic rats, *Am J Pathol*, 185 (2015) 409-419.
- [7] M. Ubeda, M. Lario, L. Munoz, M.J. Borrero, M. Rodriguez-Serrano, A.M. Sanchez-Diaz, R. Del Campo, L. Lledo, O. Pastor, L. Garcia-Bermejo, D. Diaz, M. Alvarez-Mon, A. Albillos, Obeticholic acid reduces bacterial translocation and inhibits intestinal inflammation in cirrhotic rats, *J Hepatol*, 64 (2016) 1049-1057.

REVIEWERS' COMMENTS:

Reviewer #1 (Remarks to the Author):

The authors have addressed the comments clearly. The quality of the manuscript has been greatly improved. I have no further comments on the current version.

Reviewer #2 (Remarks to the Author):

The author has appropriately addressed all concerns.